# Direct Mechanical Thrombectomy vs. Bridging Therapy in Stroke Patients in A “Stroke Belt” Region of Southern Europe

**DOI:** 10.3390/jpm13030440

**Published:** 2023-02-28

**Authors:** Cristina del Toro-Pérez, Laura Amaya-Pascasio, Eva Guevara-Sánchez, María Luisa Ruiz-Franco, Antonio Arjona-Padillo, Patricia Martínez-Sánchez

**Affiliations:** 1Stroke Centre, Department of Neurology, Torrecárdenas University Hospital, University of Almería, 04009 Almería, Spain; 2Faculty of Health Sciences, CEINSA (Center of Health Research), University of Almería, 04120 Almería, Spain

**Keywords:** direct mechanical thrombectomy, bridging therapy, intravenous thrombolysis, acute ischemic stroke, stroke belt, transfer model

## Abstract

The aim of this 4-year observational study is to analyze the outcomes of stroke patients treated with direct mechanical thrombectomy (dMT) compared to bridging therapy (BT) (intravenous thrombolysis [IVT] + BT) based on 3-month outcomes, in real clinical practice in the "Stroke Belt" of Southern Europe. In total, 300 patients were included (41.3% dMT and 58.6% BT). The frequency of direct referral to the stroke center was similar in the dMT and BT group, whereas the time from onset to groin was longer in the BT group (median 210 [IQR 160–303] vs. 399 [IQR 225–675], *p* = 0.001). Successful recanalization (TICI 2b-3) and hemorrhagic transformation were similar in both groups. The BT group more frequently showed excellent outcomes at 3 months (32.4% vs. 15.4%, *p* = 0.004). Multivariate analysis showed that BT was independently associated with excellent outcomes (OR 2.7. 95% CI,1.2–5.9, *p* = 0.02) and lower mortality (OR 0.36. 95% CI 0.16–0.82, *p* = 015). Conclusions: Compared with dMT, BT was associated with excellent functional outcomes and lower 3-month mortality in this real-world clinical practice study conducted in a region belonging to the “Stroke Belt” of Southern Europe. Given the disparity of results on the benefit of BT in the current evidence, it is of vital importance to analyze the convenience of its use in each health area.

## 1. Introduction

Intravenous thrombolysis (IVT) in eligible patients with acute ischemic stroke (AIS) and large vessel occlusion (LVO) is currently recommended before mechanical thrombectomy (MT) by American and European guidelines [1]. However, it is currently under debate whether combined therapy with IVT prior to MT, known as bridging therapy (BT), could improve the radiological results and functional outcome of patients with AIS and LVO. While IVT could favor the clot softening or achieve complete recanalization in 7% to 26% of cases [2,3,4], there is some concern about the possible increased risk of brain hemorrhage, delayed puncture time, and the presence of complications related to clot fragmentation leading to distal embolization [2,4,5,6].

In the recent years, observational studies, randomized clinical trials (RCT), and various meta-analyses have been carried out in order to determine if direct MT (dMT) may be more beneficial than BT in AIS with LVO [6,7,8,9,10]. The RCTs have shown contradictory results, some showing the non-inferiority of dMT compared to BT and others failing to demonstrate this [7,9,10,11]. Some meta-analyses including only RCTs showed a better outcome with BT [8,12,13]. However, those including observational, real-word studies have shown no significant differences in functional independence, mortality, recanalization rate, or incidence of symptomatic intracranial hemorrhage between both modalities of treatment in AIS and anterior LVO [3,7,14,15]. This heterogeneity between studies is favored by differences in the clinical characteristics of patients and strokes subtypes, as well as geographic and organizational differences between hospitals and regions. 

In Europe, the burden of vascular risk factors, cerebrovascular disease mortality, and stroke management protocols differ among the different areas. In Spain, situated in the south of Europe, there is a specific southern region of the country (regions of Andalusia and Murcia) where higher rates of cardiovascular factors and stroke mortality have been reported, called the “Stroke Belt” [15,16,17]. Similarly, Andalusia is a region with a large degree of geographical dispersion, where a high percentage of patients are initially referred to a hospital without a neurologist and, if needed, subsequently transferred to a stroke center (SC), according to a drip-and-ship model. A greater benefit of IVT has been reported in this transfer model versus direct transfer to the SC (mothership model) [2,18].

The main objective of the present real-world study is to compare the functional outcomes, clinical characteristics, and the complication rate of AIS patients who underwent dMT vs. BT in Spain’s Stroke Belt, a region with a particularly high incidence of stroke and a mixed mothership and drip-and-ship-model for MT. 

## 2. Materials and Methods

### 2.1. Patient Population

We retrospectively analyzed through the records of patients ≥ 18 years admitted to the SC with AIS in the anterior or posterior circulation and LVO who were treated with dMT or BT, according to current International Guidelines, in Almeria, province of Andalusia (Southern Spain), between February 2017 and February 2021. Almería has a referral SC, the Torrecárdenas University Hospital, serving a total population of 739,293 inhabitants. All the patients admitted to the SC benefiting from a mechanical thrombectomy were enrolled consecutively during the study period. 

AIS from the SC health area were transferred directly, following a mothership model, while those residing in other areas (approximately 60% of the population) were initially treated in one of the regional hospitals of the province, following a drip-and-ship model. From March 2021, all patients in the province switched to a mixed model depending on the stroke severity, with a mothership model if the RACE scale was greater than or equal to 5. The nearest regional hospital is 36 km away and the farthest regional hospital is 113 km from the SC. 

The variables included were: demographic data, vascular risk factors and comorbidities, previous treatments, AIS blood biomarkers, stroke characteristics and etiology according to the TOAST classification, NIHSS on admission and after 24 h, reperfusion treatments (MT and IVT), time periods (onset-to-door, onset-to-groin, and door-to-groin times), recanalization by TICI grades, number of passes during MT, in-hospital complications, and 3-month outcomes according to the modified Rankin Scale score (mRS).

### 2.2. Outcome Parameters 

The primary outcome was excellent functional outcome at 3 months, defined as a mRS score of 0–1. Secondary outcomes were death due to any cause at 3 months, defined as a mRS of 6, hemorrhagic transformation during the first 36 hours (parenchymal hematoma type 1 -PH1- of type 2 -PH2-) according to the ECASS classification, and proportion of patients with successful recanalization (TICI 2b–3) [19].

### 2.3. Data Analysis

The statistical analysis was performed using IBM SPSS Statistics for Windows, Version 25.0, software (IBM Corp. Armonk, NY, USA) for Windows. Continuous variables were expressed as mean SD or median (interquartile range [IQR]) and compared with Student’s *t*-test or the Mann–Whitney test, as appropriate. Comparisons between groups were analyzed with the two or Fisher’s exact test for dichotomous variables. The relationship between BT and 3-month excellent outcome was assessed using a multivariate binary logistic regression model. Those variables with *p* < 0.2 in the bivariate analysis performed according to prognosis were included in the multivariate analysis. Those variables that differed significantly in the bivariate analysis between treatment groups were also included. A backward procedure was followed as the modelling strategy, using the log likelihood ratiotest to assess the goodness of fit and compare nested models. Those variables that when eliminated produced a change of ≥15% in the odds ratios (OR) were considered confounding variables. The OR and the 95% confidence intervals (CI) were used to evaluate the association strength. A *p*-value of < 0.05 was considered statistically significant.

### 2.4. Ethical Issues 

The study protocol is in accordance with the ethical guidelines of the 1975 Declaration of Helsinki, including all subsequent amendments. The project was approved by the Clinical Research Ethics Committee of the Torrecárdenas University Hospital. The data collected for the study were processed in accordance with the General Data Protection Regulation (EU) 2016/679 of the European Parliament and of the Council of 27 April 2016.

## 3. Results

A total of 300 patients were included in the study, median age (IQR) 71 (61–80) years, 114 (48%) women. Overall, 139 (41.3%) were treated with dMT and 176 (58.7%) received BT. Follow-up at 3 months was not achieved in 21 patients due to transfer to the usual home in a different region, or other impediments. 

Table 1 shows the baseline and stroke characteristics. Demographic data and comorbidities were similar in both groups. Among the included patients, the median NIHSS score at baseline was 17 [12,13,14,15,16,17,18,19,20,21] and the most frequently affected territory was the anterior circulation (87.3%). Cardioembolic etiology, found in 51% of the samples, was the most prevalent. The percentage of patients undergoing a drip-and-ship model before MT was 58.3%, similar between groups. 

A higher proportion of patients with anticoagulant therapy was found in the dMT group (5.6% in BT vs. 24.4% in dMT, *p* < 0.001) as well as a higher percentage of wake-up stroke (1.7% in BT vs. 43.9% in dMT, *p* < 0.001). Similarly, the dMT group had a lower percentage of previous excellent mRS, a longer duration of stroke symptoms at the time of MT (399 minutes vs. 210 minutes, *p* < 0.001), a higher rate of awakening stroke and anticoagulation (43.9% vs. 1.7 %, *p* < 0.001; 24.4 vs. 5.6%, *p* < 0.001), and higher door-to-groin time (85 minutes in BT group vs. 97 minutes in dMT group, *p* = 0.005), with a lower median ASPECTS score (9 vs. 10, *p* < 0.001). The main reason for dMT was a time window from stroke onset above 4.5 hours. Other reasons were neoplastic disease and recent anticoagulation intake. 

The dMT group had greater rates of recanalization failure (TICI-0) compared to BT (1.7% vs. 6.7%, *p* = 0.02), with no differences in TICI 2b/3 (84.9% BT vs. 82.8% dMT. *p* = 0.63). Table 2 shows the main features of the procedure, complications, and evolution during hospitalization. 

At the 3-month follow-up, the percentage of patients who obtained an excellent functional outcome was twice as high in the patients who received BT (32.4 % BT vs. 15.4% dMT, *p* = 0.004), without a greater rate of symptomatic bleeding between the two groups (16.4% BT vs. 16.2% dMT, *p* = 0.98). Mortality at 3 months was almost twice as high in the dMT group (14.2% BT vs. 26% dMT). Figure 1 shows the 3-month prognosis assessed by the mRS in both treatment arms. 

The multivariate logistic regression analysis (Table 3) showed that BT was independently associated with an excellent 3-month outcome (OR 2.7. 95% IC, 1.2–5.9. *p* = 0.017) adjusted by confounders. Higher age (OR 0.96. 95% IC, 0.93–0.99. *p* = 0.007) and NIHSS score (OR 0.86. 95% IC, 0.81–0.92. *p* < 0.001), prior anticoagulation (OR 0.28. 95% IC, 0.09–0.89. *p* = 0.032), significant cerebral edema (OR 0.12. 95% IC, 0.025–0.6. *p* = 0.01), and lower respiratory tract infection during hospitalization (OR 0.1. 95% IC, 0.025–0.42) were inversely associated with a mRS 0–1 at three months. Three or fewer passes during the MT was independently associated with an excellent outcome (OR 5.3. 95% IC, 1.79–15.56. *p* = 0.002). 

Another multivariate logistic regression analysis was performed considering death in the first three months, showing BT to be a protective factor (OR 0.36. 95% CI 0.16–0.82, *p* = 015). Higher age (OR 1.04. 95% CI 1.01–1.09. *p* = 0.016) and NIHSS (OR 1.09. 95% CI 1.02–1.17, *p* = 0.012), lower respiratory tract infection during hospitalization (OR 3.89. 95% CI 1.58–9.56, *p* = 0.003), and use of antiplatelet agents (OR 0.4 % CI 0.67–0.96. *p* = 0.04) were also associated with death. 

## 4. Discussion

To the best of our knowledge, this is the first study that analyzes whether BT, compared with dMT, improves stroke outcomes in clinical practice in a region belonging to the Southern European “Stroke Belt”. In the BT group, we observed a three-fold higher rate of patients achieving an excellent outcome (mRS 0–1) at three months after adjustment for confounding variables, as well as lower mortality. This higher rate of excellent prognosis was not associated with a higher rate of significant hemorrhagic transformation.

Previous RCTs have yielded conflicting evidence about the benefit of IVT prior to MT. While two large RCTs showed non-inferiority of dMT versus BT with no significant differences in symptomatic intracerebral hemorrhage, a third Asian RCT as well as three other European RCTs did not [7,9,10,11,21,22,23]. Except for one, all of them included anterior circulation strokes exclusively [22]. The two RCTs that favored dMT were performed in Asian populations where there is a higher frequency of stroke of atherothrombotic etiology, in which IVT has been proven to be less effective than in cardioembolic strokes [24]. This also implies that this population may require endovascular stenting during MT, requiring more intensive antiplatelet therapy which might increase the risk of intracranial hemorrhage when combined with IVT. In fact, one study observed a slightly higher rate of symptomatic intracranial bleeding in Asian patients treated with IVT when compared with non-Asian patients including white, black, and Hispanic patients [25]. Despite this, a recent large RCT involving Asian and European patients failed to prove non-inferiority of dMT, showing an even greater benefit from BT when the Asian population subgroup was analyzed [22].

When this question is analyzed in real-world studies, the balance tips toward a greater benefit of BT treatment regarding the 90-day functional outcome [5,26,27,28,29,30,31], although some heterogeneity persists with some authors reporting no significant differences between the two groups [32,33].

Likewise, several meta-analyses have been conducted in order to find an answer regarding the best therapeutic option in AIS with LVO. Those that have benefited the BT group included mainly observational, real-world, non-randomized studies, showing a better functional outcome, higher recanalization rates, and lower mortality with BT [8,12,34,35]. This data is consistent with those presented in this study. However, many limitations are evident in these sorts of studies, including the lack of homogeneity between treatment groups regarding age, NIHSS at admission, and time from stroke onset, which restrain the generalization of the results and their degree of evidence [6,8,20]. In our case, the baseline characteristics between the two groups differed in the percentage of patients with excellent mRS before the stroke, the onset-to-groin time, rate of anticoagulation treatment, awakening strokes, and the median ASPECTS score. However, in the multivariate analysis, these variables did not change the results in terms of BT benefit. In contrast to these, some meta-analyses that also included a majority of non-randomized studies have not supported the use of BT [14]. Liu et al. and Kaesmacher et al. observed no differences between groups in the rate of successful recanalization, but reported a trend toward a lower recanalization rate in dMT patients in whom IVT could not be performed for clinical reasons and a trend towards a higher recanalization rate in dMT patients that were IVT-eligible patients, suggesting that among IVT-eligible patients, dMT may be associated with equal rates of angiographic results [14,20]. This may be explained by the comorbidities of patients in whom IVT was contraindicated [6,14,20]. Despite this, other meta-analyses have not shown a significant difference in recanalization even if only dMT patients eligible for IVT vs. BT were compared [20]. Through subgroup analyses in different meta-analyses, a worse prognosis has been observed in the dMT group when both IVT-subsidized and non-IVT-subsidized patients were included. In analyses in which only IVT-subsidized patients were included in the dMT group, the analyses show similar prognosis and mortality [20,35]. Again, this is consistent with the results of observational studies such as ours, in which patients with dMT may or may not be subsidiary to receiving IVT. This implies that the BT and dMT groups have different baseline characteristics, such as a higher percentage of patients treated with anticoagulation and a longer time of stroke evolution at the time of the procedure, although in our sample these two factors were not related to 3-month outcomes. Although it has been speculated that IVT prior to MT may imply a higher risk of cerebral hemorrhage, delay in puncture time and the presence of complications related to clot fragmentation leading to distal embolization [2,4,5,6], meta-analyses have not shown a higher percentage of hemorrhage or delay in door-to-groin time, findings which are replicated in our sample. On the contrary, IVT could be beneficial by favoring clot softening. It could also be of greater benefit in multifocal ischemia or in cases of multiple distal occlusions, by increasing the rates of successful recanalization and even avoiding the need for MT [1,14,15].

Several factors explain this heterogeneity between studies, such as differences in the clinical and ethnic-related characteristics of the population, stroke severity, recanalization rates according to the size and location of the occluded artery (anterior or posterior), the transfer model (mothership or drip-and-ship), stroke etiology, and resources in the health area [2,18,36,37]. BT appears to be superior to dMT in mild stroke, with no benefit of prior IVT with higher NIHSS scores [36]. This is probably due to the likely association of elevated NIHSS with major vessel occlusion, as the efficacy of IVT is more than three times higher in occlusion of the M2 segment of the middle cerebral artery compared to the internal carotid artery [38]. Likewise, a score below 6 points on the ASPECTS scale has also been related to the non-benefit of IVT prior to MT, with BT being beneficial over dMT in those patients with an ASPECTS score equal to or greater than 6 [39]. Nie et al. reported better results with BT in patients with basilar artery occlusions [40]. In two studies, the efficacy of BT was evaluated in patients with a cardioembolic stroke, showing no differences and a worse functional outcome in the BT group, respectively [41,42]. There is also no consensus regarding tandem carotid occlusions, with some data supporting a better prognosis with BT and others showing a higher mortality in this group [32,38]. In the present analysis, no differences in outcome were identified according to the stroke etiology or tandem occlusions.

Most of the previous studies that have addressed whether IVT prior to MT improves the stroke outcome have not analyzed the results based on the patient transfer model [1,2,5,6,7,14,15,43]. This is of significant importance when generalizing the net benefits of MT alone as well as the benefit of prior IVT. The frequent need for interhospital transfers implies that the results from a given area cannot be extrapolated to other healthcare areas, with each area having to perform its own analysis to determine the suitability of one treatment or another (BT vs. dMT), as well as the benefit of one transport model versus another (mothership vs. drip-and-ship). The efficacy of BT has been found to be substantially higher in patients with a secondary transfer to the SC where MT is performed (drip-and-ship model) compared to patients who underwent BT at the SC, without interhospital transfers (mothership model) [2,18]. This difference between transfer models could be explained due to longer exposure time to tPA [37]. Our analysis is consistent with this fact, as it was performed on a sample that included a drip-and-ship model. A recent study conducted in the province of Catalonia, in the north of Spain, showed no differences in prognosis according to the transport model [44]. However, it is an area with a different organization and geographical dispersion to Andalusia and has different health areas, so the most appropriate transfer model should also be individualized in each of them. Taking into account the transfer model and homogenizing the sample according to stroke characteristics when analyzing the benefit of IVT in LVO could help to reduce the heterogeneity between studies and future meta-analyses. 

It is worth mentioning that this study has been carried out in an area within the so-called “Stroke Belt”, performed for the first time in this population group with specific characteristics and cardiovascular risk factors. In studies carried out in the 2010s, it was observed that Andalusia had higher rates of risk factors such as hypertension, diabetes, cardiovascular disease, overweight, obesity, and smoking, as well as worse socioeconomic data (education and income) [15,16]. It takes approximately 40% of the population more than an hour and a half to obtain care from a Neurology specialist. Recently, the introduction of a centralized telestroke network has allowed this subgroup of patients to access this specialized care in less than 30 minutes. Despite this, transfer to the Thrombectomy Center continues to be required and a drip-and-ship model of organization predominates [45]. This is especially relevant given the greater benefit of IVT in this subgroup of patients [2,18].

On the other hand, a future has been recently opened with the administration of intra-arterial fibrinolysis after the performance of MT, following recent results published by Renú. A. et al. [46]. The introduction of tenecteplase as a thrombolytic agent will also be a game-changer in the BT debate [1,47,48]. In any case, the current evidence shows that BT with alteplase does not lead to a higher percentage of hemorrhages, nor does it delay the onset of MT.

The main limitation of our study is its retrospective nature, which limited or prevented complete follow-up of some patients. In addition, the treatment decision was based on clinical characteristics with the aforementioned differences between groups. 

## 5. Conclusions

Compared to dMT, BT was associated with an excellent functional outcome and lower mortality at 3 months in a study of real-world clinical practice in a region belonging to the “Stroke Belt” of Southern Europe. Given the previously reported disparity of findings on the benefit of IVT prior to MT in ischemic stroke patients with LVO, it is of vital importance to analyze the convenience of its use in each health area, given that the demographic characteristics, the facilities of the regional hospitals, the health organization, and the transport times to the thrombectomy center differ from one community to another. 

## Figures and Tables

**Figure 1 jpm-13-00440-f001:**
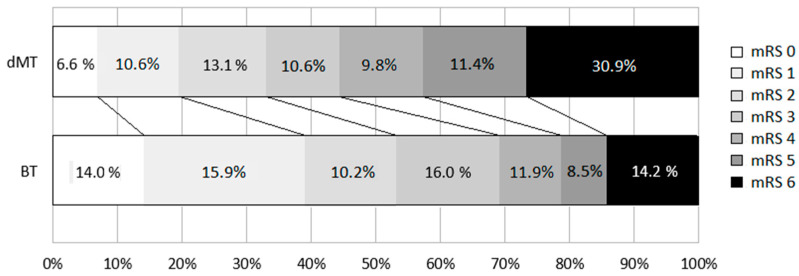
Modified Rankin Scale distribution at 90 days *p* < 0.001 for comparison between groups. dMT, direct mechanical thrombectomy (dMT); BT, bridging therapy. Loss to follow-up resulted in the loss of data regarding 3-month prognosis based on mRS, which was analyzed in a total of 279 patients.

**Table 1 jpm-13-00440-t001:** Baseline and stroke characteristics.

Variable	Total (N = 300)	BT (N = 176)	dMT (N = 124)	*p*
**Demographic data and comorbidities**				
Female, n (%)	114 (38)	77 (43.8)	46 (37.4)	0.27
Age, median (IQR), years	71 (19)	71 (20)	74 (19)	0.061
Previous mRS ≤ 1	257 (85.7)	156 (88.7)	101 (81.4)	0.02
Smoker, n (%)	55 (18.3)	37 (21)	26 (21.1)	0.72
Alcoholism, n (%)	28 (9.3)	20 (11.4)	14 (11.4)	0.66
Drugs, n (%)	5 (1.6)	2 (1.1)	3 (2.4)	0.37
Glycemia (IQR), mg/dL	127 (52.5)	127 (58)	126.5 (49.3)	0.76
Hypertension, n (%)	173 (57.6)	105 (59.7)	84 (68.3)	0.15
Dyslipidemia, n (%)	121 (49.3)	78 (44.3)	53 (43.1)	0.8
Diabetes mellitus, n (%)	83 (27.7)	46 (26.1)	40 (32.5)	0.23
Ischemic cardiopathy, n (%)	43 (14.3)	25 (14.2)	20 (16.3)	0.6
Stroke, n (%)	26 (8.7)	7 (3.9)	21 (17.1)	0.4
Previous atrial fibrillation, n (%)	64 (21.3)	33 (18.8)	35 (28.5)	0.08
De novo atrial fibrillation *, n (%)	74 (24.7)	50 (28.4)	30 (24.4)	0.28
**Prior treatments**				
Statins, n (%)	127 (42.3)	81 (46)	52 (42.3)	0.51
Antiplatelet agents, n (%)	97 (32.3)	55 (31.3)	42 (34.2)	0.93
Anticoagulants, n (%)	37 (12.3)	10 (5.6)	30 (24.4)	<0.001
**Underdosed Vitamin K antagonist **, n (%)**	19 (6.3)	5 (2.8)	14 (11.4)	0.015
**Direct referral to Stroke Centre**	175 (58.3)	112 (63.6)	75 (60.9)	0.65
**Vascular territory**				
Anterior circulation, n (%)	262 (87.3)	157 (89.2)	105 (85.4)	0.16
Left side, n (%)	146 (48.7)	84 (47.4)	62 (50.4)	0.22
**NIHSS (median, IQR)**	17 (9)	16 (8)	18 (9)	0.27
**ASPECTS (median, IQR)**	10 (2)	10 (1)	9 (3)	<0.001
**Awakening stroke, n (%)**	50 (16.7)	3 (1.7)	54 (43.9)	<0.001
**Stroke etiology (TOAST)**				0.15
Cardioembolic, n (%)	153 (51)	94 (53.4)	70 (56.9)	
Atherothrombotic, n (%)	53 (17.6)	37 (21)	21 (17.1)	
Other determined etiology, n (%)	14 (4.7)	3 (1.7)	11 (8.9)	
Undetermined etiology, n (%)	51 (17)	39 (22.1)	19 (15.4)	
**Blood biomarkers**				
Platelet volume ≥ 9.6 fL n (%)	102 (34)	67 (38)	35 (28.5)	0.18
NT-proBNP (IQR), pg/mL	1154 (2275.3)	1153.5 (2195.5)	1182 (2540.6)	0.76

* Diagnosed on admission or during hospitalization. ** Defined as INR < 1.7. BT, bridging therapy; dMT, direct mechanical thrombectomy; NIHSS, National Institutes of Health Stroke Scale; TOAST, Trial of Org 10,172 in Acute Stroke Treatment classification; NT-proBNP, N-terminal pro-brain natriuretic peptide.

**Table 2 jpm-13-00440-t002:** Main features of the procedure, in-hospital complications, and 3-month outcomes.

Variable	Total (N = 300)	BT (N = 176)	dMT (N = 124)	*p*
**Procedure**				
Onset-to-groin time, median (IQR), min	245 (222)	210 (143)	399 (450)	<0.001
Door-to-groin time (IQR), min	88 (46.5)	85 (40)	97 (70)	0.005
MT duration, median (IQR), min	32 (40)	31 (40)	32 (49)	0.44
Passes in MT ≤ 3, n (%)	47 (15.7)	96 (54,5)	72 (58.1)	0.64
Stent implantation, n (%)	45 (15)	23 (13.1)	22 (17.9)	0.26
TICI 2b-3	196 (65.3)	122 (69.3)	74 (60.2)	0.57
TICI 0	11 (3.7)	3 (1.7)	8 (6.5)	0.02
**In-hospital complications**				
Hemorrhagic transformation *, n (%)	49 (16.3)	29 (16.4)	20 (16.2)	0.98
Brain edema with midline deviation, n (%)	67 (22.3)	40 (22.7)	27 (21.9)	0.93
Craniectomy, n (%)	8 (2.7)	6 (3.4)	2 (1.6)	0.28
Renal failure, n (%)	37 (12.3)	25 (14.2)	15 (12.2)	0.58
Lower respiratory tract infection, n (%)	64 (21.3)	36 (20.5)	32 (26)	0.27
Urinary tract infection, n (%)	23 (7.7)	16 (9.1)	10 (8.1)	0.76
**3-month outcomes**				
Excellent outcomes (mRS 0–1), n (%)	76 (25.3)	57 (32.4)	19 (15.4)	0.004
Death, n (%)	57 (19)	25 (14.2)	32 (26)	0.002

* Parenchymal hematoma type 1 (PH1) of type 2 (PH2) according to the ECASS classification. BT, bridging therapy; dMT, direct mechanical thrombectomy; MT, mechanical thrombectomy; TICI, thrombolysis in cerebral infarction scale score; NIHSS, National Institutes of Health Stroke Scale; mRS, modified Rankin Scale score.

**Table 3 jpm-13-00440-t003:** Multivariate analysis of factors associated with 3-month excellent outcome (mRS 0–1).

Variable	Unadjusted Analysis		Adjusted Analysis *	
	OR (95% CI)	*p*	OR (95% CI)	*p*
**Demographic data and comorbidities**				
Age, years	0.86 (0.45–1.6)	0.001	0.96 (0.94–0.99)	0.007
Previous mRS ≤ 1	2.73 (1.15–5.57)	0.02	-	-
HTA	0.71 (0.43–1.1)	0.15	-	-
FA	0.61 (0.36–1.06)	0.08	-	-
Anterior circulation	0.59 (0.27–1.3)	0.16	-	-
**Prior treatments**			-	-
Anticoagulants	0.19 (0.092–0.42)	<0.001	0.28 (0.09–0.85)	0.032
**NIHSS**	0.88 (0.85–0.93)	<0.001	0.86 (0.81–0.92)	<0.001
**ASPECTS**	1.2 (0.98–1.4)	0.08	-	-
**Awakening stroke**	0.022 (0.007–0.074)	<0.001	-	-
**Stroke etiology (TOAST)**			-	-
Cardioembolic (reference)	Reference	0.03	-	--
Atherothrombotic	1.31 (0.71–2.43)	0.39	-	-
Other determined etiology	0.20 (0.06–0.76)	0.02	-	-
Undetermined etiology,	1.52 (0.81–2.87)	0.19	-	-
**Serum biomarkers**				
NT-proBNP, pg/mL	1.00 (1.00–1.00)	<0.001	-	-
**Acute treatment**				
BT	2.03 (1.2–3.6)	0.013	2.67 (1.2–5.9)	0.017
**Procedure**				
Onset-to-groin time, minutes	1.00 (0.99–1.001)	0.424	-	-
Door-to-groin time, minutes	0.99 (0.99–1.00)	0.003	-	-
MT duration, min	0.99 (0.98–1.01)	0.07	-	-
Passes in MT ≤ 3	4.1 (1.6–10.2)	0.001	5.29 (1.79–15.56)	0.002
TICI 2b-3	2.8 (1.2–7.1)	0.03	-	-
TICI 0	0 (0)	0.99	-	-
**In-hospital outcomes and complications**				
Brain edema with midline deviation	0.061 (0.014–0.26)	<0.001	0.12 (0.025–0.6)	0.01
Hemorrhagic transformation	0.1 (0.02–0.4)	<0.001	-	-
Craniectomy	0.72 (0.67–0.78)	0.081	-	-
Lower respiratory tract infection	0.09 (0.03–0.32)	<0.001	0.1 (0.025–0.42)	0.002

* Adjusted by age, previous mRS ≤ 1, HTA, FA, anterior circulation, prior anticoagulants, NIHSS, ASPECTS, rate of awakening stroke, stroke etiology, NT proBNP, BT, onset-to-groin time, door-to-groin time, MT duration, number of passes in MT, TICI 2b-3, TICI 0, brain oedema with midline deviation, hemorrhagic transformation, craniectomy, respiratory infection; mRS, modified Rankin Scale score; BT, bridging therapy; MT, mechanical thrombectomy; TICI, thrombolysis in cerebral infarction scale score; NIHSS, National Institutes of Health Stroke Scale; ASPECTS, Alberta Stroke Programme Early CT Score.

## Data Availability

The data presented in this study are available upon request from the corresponding author. Data are not available to the public due to personal data protection.

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
