# Peer review of "Direct Mechanical Thrombectomy vs. Bridging Therapy in Stroke Patients in A “Stroke Belt” Region of Southern Europe"

_jpm, 2023, doi:10.3390/jpm13030440_

Round 1
Reviewer 1 Report
The authors presented the manuscript regarding the use of direct MT vs. bridging therapy (IVt + MT) for stroke patients.
Given the significance of the content and the results of the study regarding development and organization of stroke network in Europe I would like to suggest the Editors to accept this manuscript.
I encourage the authors to perform the multi-centre analysis for more accurate and precise results in the future.
Author Response
The recommendation of a future multi-centre analysis had been considered as next-step research in this field.
Reviewer 2 Report
As the authors noted, the benefits of BT vs. dMT as well as mothership vs. drip-and-ship models of care may depend on regional circumstances and should be assessed locally. Therefore, from my perspective, the study sets a valid example of this approach. In other countries or regions (e.g., https://www.ncbi.nlm.nih.gov/pmc/articles/PMC9726865/) the outcomes could differ. The authors could mention the findings of the above-mentioned Egyptian study in the Discussion to make this point.
Author Response
In agreement with the reviewer, we add the quote with the rest of real world studies on line 234.
Reviewer 3 Report
This retrospective and single-center study aimed to investigate the outcomes of stroke patients treated with direct mechanical thrombectomy (dMT) compared to bridging therapy (BT) (intravenous thrombolysis (IVT) + MT) based on 3-month outcomes, in real-world clinical practice in southern European “Stroke Belt”. This study found that the BT group more frequently showed an excellent 3-month outcome. Multivariate analysis also showed that BT was independently associated with excellent outcome and lower mortality. The authors proposed that given the disparity of results on the benefit of BT in the current evidence, it is of vital importance to analyze the convenience of its use in each health area. This paper was well written using appropriate statistical methodology. However, approval requires the following issues to be addressed:
1. DIRECT MECHANICAL THROMBECTOMY VERSUS BRIDGING THERAPY FOR STROKE PATIENTS IN A SOUTHERN EUROPEAN "STROKE BELT". From its title, this study is reminiscent of a multi-center study conducted in several regions in southern Europe. However, contrary to expectations, this is a single-institutional study. Therefore, more appropriate research titles should be considered.
2. How were the 300 patients extracted? Did the authors enroll patients consecutively from 2017 to 2021? Or did the authors select 300 patients using a statistical matching method among patients hospitalized during the study period? Please clarify these issues by presenting a patient engagement flow chart as a figure.
3. Why was TICI 0, which showed a statistically significant difference between groups, not adjusted in multivariate analysis?
4. Does a similar multivariate analysis result come out even if only the statistically significant variables in Tables 1 and 2 are adjusted?
5. There are some typing errors.
Author Response
1.The authors agree with the reviewer´s suggestion, therefore a new title has been proposed.
2. All the patients admitted to the stroke centre benefiting from a mechanical thrombectomy were enrolled consecutively during the study period. We add an explanatory sentence in methodology
3.As suggested by the reviewer, we explored the effect of TICI-0 on patients' functional outcome (excellent vs. non-excellent rankin at 90 days), without finding a significant effect in the bivariate logistic regression analysis. This finding has been added to table 3.
4. Table 1 represents the differences between the two treatment groups. Regarding the functional outcome, when we perform the multivariate analysis including only statistically significant variables (p < 0.05 instead of p < 0.2), the same variables are retained by the model.
5. The manuscript has been reviewed in order to correct them
Round 2
Reviewer 3 Report
Thanks.